# The neutrophil-to-lymphocyte ratio is associated with coronary heart disease risk in adults: A population-based study

Yu Wang[1,2,3,4,5☯], Yangping Zhuang[1,2,3,4,5☯], Changsha Lin[6☯], Hanqing Hong[6], Feng Chen[1,2,3,4,5]*, Jun Ke[1,2,3,4,5]*

**1** Shengli Clinical Medical College of Fujian Medical University, Fuzhou, China, **2** Department of Emergency, Fujian Provincial Hospital, Fuzhou, China, **3** Fujian Key Laboratory of Emergency Medicine, Fujian Provincial Hospital, Fuzhou, China, **4** Fujian Provincial Institute of Emergency Medicine, Fuzhou, China, **5** Fujian Emergency Medical Center, Fuzhou, China, **6** Jinan Branch of Jinjiang City Hospital, Jinjiang, China

☯ These authors contributed equally to this work.
* cf9066@126.com (FC); 68223384@qq.com (JK)

**Data Availability Statement:** All NHANES data for this study are publicly available and can be found here: https://wwwn.cdc.gov/nchs/nhanes.

## Abstract

The purpose of this study was to look at any connections that could exist between neutrophil-lymphocyte ratio and coronary heart disease. We performed a cross-sectional research of 13732 participants in the National Health and Nutrition Examination Survey who were 40 or older. Multivariate logistic regression models investigated the relationship between neutrophil-to-lymphocyte ratio levels and coronary heart disease risk. To investigate potential nonlinear connections, smoothed curve fitting was used. When a nonlinear relationship was discovered, the inflexion point was determined using a recursive method. After controlling for relevant confounders, neutrophil-to-lymphocyte ratio was independently linked to a higher risk of coronary heart disease (OR = 1.74, 95% CI:1.30–2.33, **P** = 0.0002). Subgroup analyses showed statistically significant positive associations between neutrophil-to-lymphocyte ratio and coronary heart disease risk in women (OR = 1.25, 95% CI:1.09–1.43), participants 60 years of age and older (OR = 1.09, 95% CI:1.00–1.19), smoking status for every day or not at all (OR = 1.23, 95% CI:1.00–1.52; OR = 1.09, 95% CI:1.00–1.19), alcohol use status for moderate alcohol use (OR = 1.11, 95% CI:1.00–1.22), body mass index >30 kg/m$^2$ (OR = 1.42, 95% CI:1.10–1.82), hypertensive (OR = 1.11, 95% CI:1.02–1.22), and individuals without diabetes (OR = 1.17, 95% CI:1.06–1.31). A positive correlation between neutrophil-to-lymphocyte ratio levels and coronary heart disease risk was also seen by smoothing curve fitting, with an inflexion point of 1.08 that was statistically significant (**P**<0.05). Our research shows elevated neutrophil-to-lymphocyte ratio levels are linked to a higher risk of coronary heart disease.

## 1. Introduction

Cardiovascular disease is one of the main causes of death globally, and coronary heart disease is one of the main causes of cardiovascular disease [1, 2]. It is believed that in addition to coronary atherosclerosis and spasm, the pathological factors contributing to coronary heart disease

**Funding:** This work was supported by the Fujian Provincial Major Scientific and Technological Special Projects on Health (NO. 2022ZD01008).

**Competing interests:** The authors have declared that no competing interests exist.

include problems with lipid metabolism, damage to vascular endothelial cells, and ongoing, dynamic inflammatory processes in the vasculature [3–5]. Coronary heart disease is a dangerous condition that is perilous to people's lives and has a significant financial impact [6, 7]. Therefore, identifying coronary heart disease risk factors is of utmost clinical significance [8].

A key factor in the aetiology of cardiovascular disease is inflammation [9–11]. The inflammatory response is a significant pathophysiological factor in the development and progression of atherosclerosis [12–14]. Under inflammatory conditions, there will be corresponding changes in different types of leukocytes, such as neutrophils and lymphocytes. The relative proportions of leukocyte subtypes can reflect the immune status and inflammation of the organism and, therefore, provide a specific reference for diagnosing and treating the disease [15]. Neutrophils release inflammatory mediators that can cause non-specific inflammation and activate vessel walls [16]. In contrast, lymphocyte counts control the inflammatory response and have an anti-atherosclerotic impact. They also show whether the body is stressed or has a compromised immune system [17]. As a result, the connection between neutrophil-to-lymphocyte ratio (NLR) and coronary heart disease risk is unclear and needs more research. The neutrophil-to-lymphocyte ratio (NLR) is regarded as an indicator of inflammation, and it is important to assess the relationship between NLR levels, a biochemical indicator of inflammation that has received much attention in clinical interventions, and coronary heart disease risk [18–20]. This research may reveal novel elements involved in the development of coronary heart disease. On the other hand, it may help define the direction of future fundamental research and offer suggestions for a novel inflammatory biomarker NLR implicated in the aetiology of coronary heart disease. Adults over the age of 40 who are middle-aged or older are more prone to have coronary heart disease. To ascertain whether there is a correlation between NLR levels and coronary heart disease risk in middle-aged and older populations, this study aimed to investigate the link.

## 2. Materials and methods

### Statistical and example sources

The National Center for Health Statistics (NCHS) used the National Health and Nutrition Examination Survey (NHANES) nationwide population-based cross-sectional survey to gather data on possible health risk factors and nutritional status of the civilian noninstitutionalized population in the United States. A sophisticated stratified, multistage probability cluster sampling design was used to choose a representative sample of the whole U.S. population [21].

By participating in routine in-home assessments, health screenings at mobile screening centres, and laboratory testing to gather laboratory data, participants were evaluated for their medical and physiological state. Because these were the only survey cycles with the complete set of factors that could be used to compute NLR, we chose four NHANES cycles from 2011 to 2018 to examine the relationship between NLR and increased risk of coronary heart disease.

We did not include data in the analysis for participants under the age of 40 years or for missing full NLR or coronary heart disease. The 13732 eligible participants were included in our final analysis after 39,156 people from the four NHANES cycles from 2011 to 2018 were disregarded due to missing data for the NLR (N = 7054), coronary heart disease (N = 11,555), and age less than 40 years (N = 6815) (Fig 1).

### Ethical review

The NCHS Research Ethics Review Committee approved the NHANES study methodology. All survey participants provided informed written permission. Accessible information on the NHANES, the research's strategy and data may be found at https://www.cdc.gov/nchs/nhanes/.

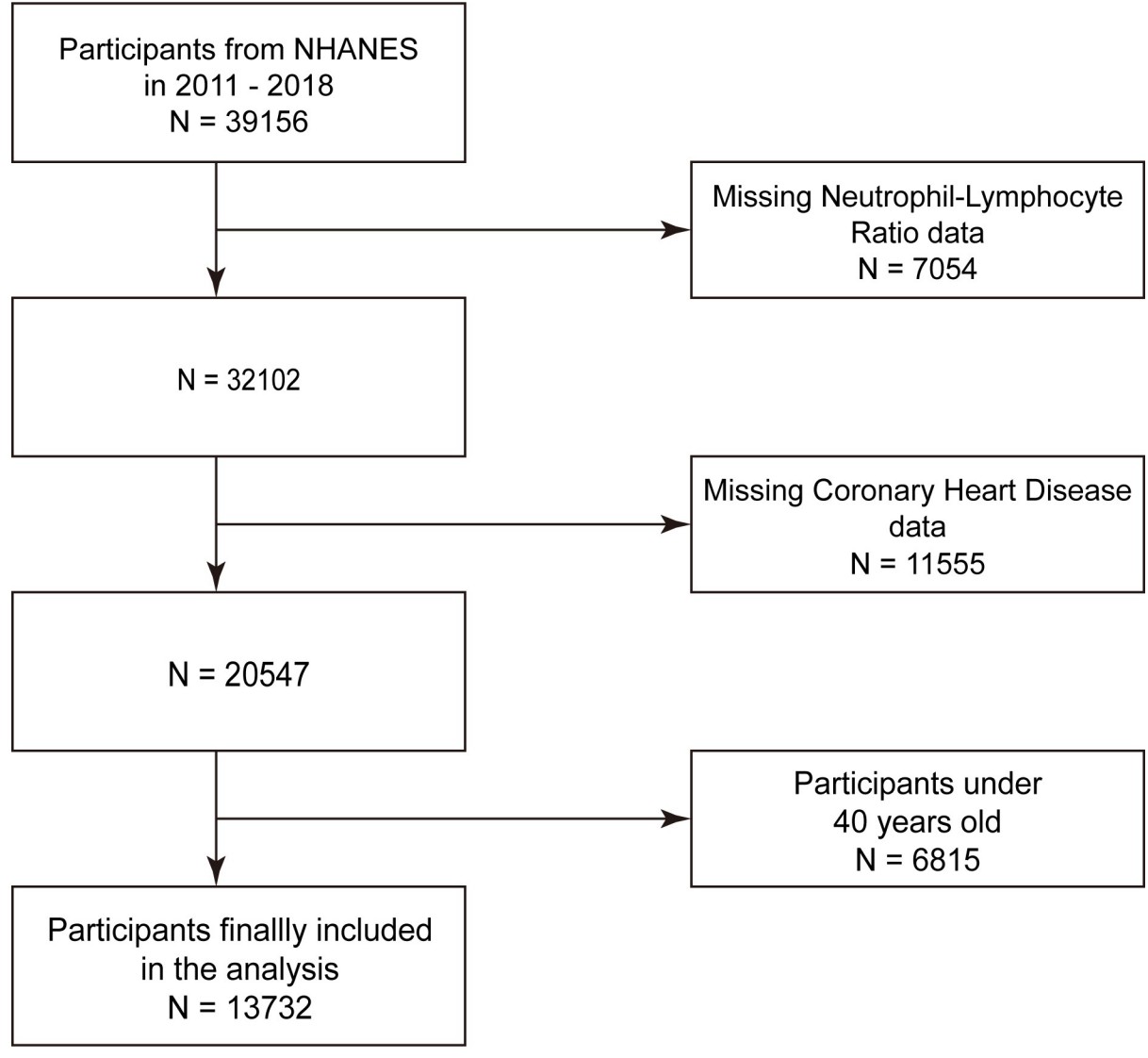

**Fig 1. Flowchart of participant selection.**

## Neutrophil-to-lymphocyte and coronary heart disease explanation

The whole blood cell counts represented as 103 cells/l, and lymphocyte and neutrophil counts were done using an automated blood analysis instrument (Coulter® DxH 800 analyzer). We estimated NLR as neutrophil /lymphocyte count, as described in earlier publications. NLR served as the exposure variable in our research.

An online questionnaire was used to evaluate the prevalence of coronary heart disease. Participants were asked, "Has a doctor or other health professional ever told you that you had coronary heart disease? " by a trained interviewer. They gave a yes/no response. They were regarded as missing if they could not respond or did not know. Coronary heart disease served as the outcome variable in our analysis.

## Covariates

Additionally, covariates such as gender (Male, Female), age (years), race (Mexican American, Other Hispanic, Non-Hispanic White, Non-Hispanic Black, Other Race), marital status (Married/Living with partner, Widowed/Divorced/Separated, Never married), education level (Less than high school, High school, More than high school), smoking status (Every day, Some days, Not at all), alcohol use status (Excessive alcohol use, Moderate alcohol use), body mass index (BMI, 0-25kg/m$^2$ normal, 25-30kg/m$^2$ overweight, >30kg/m$^2$ obese), hypertension (Yes, No), diabetes (Yes, No), serum creatinine (SCr, umol/L), serum urea nitrogen (BUN, mmol/L), serum albumin (Sal, g/L), high-density lipoprotein cholesterol (HDL-C, mmol/L), and low-density lipoprotein cholesterol (LDL-C, mmol/L) have been accounted for in our study. Information about the participants on their gender, age, race, education level, marital status, diabetes, hypertension, and smoking status and alcohol use status was gathered by trained interviewers using a computer-assisted personal interview system. A participant was questioned: "Was there ever a time or times in your life when you drank 5 (male)/4 (female) or more drinks of any kind of alcoholic beverage almost every day?", "Do you now smoke cigarettes?", "Have you ever been told by a doctor or other health professional that you had hypertension, also called high blood pressure?", "Have you ever been told by a doctor or health professional that you have diabetes or sugar diabetes?". Participants gave a yes or no response. It was referred to as missing data if they declined to respond or could not provide an answer. It was referred to as missing data if they declined to respond or could not provide an answer. By taking the height and weight of the participant into account, the BMI was determined, which was then divided into three ranges: 0-25kg/m$^2$ normal, 25-30kg/m$^2$ overweight, >30kg/m$^2$ obese. By using the Jaffe rate technique and calibrating it with standard isotope dilution mass spectrometry, SCr was determined. Using a Beckman Synchron LX20, routine biochemical analysis was used to determine LDL-C, HDL-C, SAl, and BUN. The complete measurement methods for these variables are all accessible to the public at www.cdc.gov/nchs/nhanes/.

## Statistical analysis

The Centers for Disease Control and Prevention (CDC) recommended that all statistical analyses be carried out using the proper NHANES sampling weights and accounting for detailed multistage subgroup surveys. Categorical variables are given as proportions, whereas continuous data are expressed as mean and standard deviation. Missing values for categorical variables were entered using the distribution of the available instances, while missing values for continuous variables were entered using the median. The of-group differences between persons with and without coronary heart disease and NLR gradient categories were evaluated using the Kruskal-Wallis rank sum (continuous variable) and Fisher's exact probability test (categorical variable). Using multivariate logistic regression, three alternative models were tested for the connection between NLR and coronary heart disease. Covariate adjustments were not applied in crude model at all. Gender, age, and race were modified in minimally adjusted model. Fully adjusted model included adjustments for gender, age, race, education level, marital status, BMI, SCr, SAl, HDL-C, LDL-C, hypertension, diabetes, smoking status, and alcohol use status. In addition, NLR was functionally converted (ln transform) for the regression analysis since NLR is skewed. The model used odds ratios (OR) and 95% confidence intervals (95% CI) to evaluate NLR and coronary heart disease. Gender (Male/Female), age (<60/≥60 years), smoking status (every day/some days/not at all), alcohol use status (excessive alcohol use/moderate alcohol use), BMI (0-25/25-30/>30 kg/m$^2$), hypertension (Yes/No), and diabetes (Yes/No) were the stratification factors used for the subgroup analysis of the association between NLR and coronary heart disease. To test for heterogeneity in the relationships between subgroups, interaction

effects were also incorporated, and these stratification variables were also taken into account as pre-specified possible effect modifiers. A threshold effects analysis model examined associations and inflexion points between NLR and coronary heart disease. Last, the stratification of the NLR and coronary heart disease connection according to sex, age, hypertension, and diabetes. R version 4.2.2 (http://www.R-project.org) and Empower software (www.empowerstats.com) were used for all analyses. The *P*<0.05 was chosen as the threshold for statistical significance.

## 3. Results

### Baseline participant characteristics

There were 13732 participants, 51.86% of women and 48.14% of men. The average age was 59.722±12.057 years. The concentration of Mean NLR±SD was 2.226±1.322. Coronary artery disease was identified in 5.99% of the patients.

Table 1 displays the clinical and biochemical features of the individuals by NLR tertiles. In tertile 1, tertile 2, and tertile 3, respectively, the prevalence of coronary heart disease was 4.25%, 5.03%, and 8.67%; it rose with increasing NLR. Age, sex, race, marital status, smoking status, alcohol use status, BMI, hypertension, diabetes, SCr, SAl, BUN, HDL-C, and coronary heart disease were all statistically different in all NLR tertiles (All *P*<0.05). In our study, NLR levels were elevated in the participant populations of age, male, non-Hispanic Whites, smoking status for not at all, excessive alcohol use status, BMI >30 kg/m$^2$, diabetes, SCr, BUN, coronary artery disease, and were decreased in the populations of non-Hispanic Blacks, other Races, HDL-C, and LDL-C (All *P*<0.05).

Table 2 displays the participant baseline characteristics broken down by the coronary heart disease status column stratification variable. Age, sex, race, marital status, smoking status, alcohol use status, diabetes, hypertension, SCr, SAl, BUN, HDL-C, and NLR all had statistically significant associations with the prevalence of coronary heart disease (*P*<0.05).

### NLR and coronary heart disease risk each other

According to our findings, a greater NLR is linked to a higher chance of developing coronary heart disease(Table 3). Both our crude model (OR = 2.15; 95% CI:1.87–2.48, *P*<0.0001) and the model with the minimally adjusted model (OR = 1.45; 95% CI:1.25–1.69, *P*<0.0001) found this connection to be significant. The fully adjusted model did not change the positive link between NLR and coronary heart disease (OR = 1.74; 95% CI:1.30–2.33, *P* = 0.0002), showing that the risk of coronary heart disease increased with each unit higher of ln-NLR. Using the tertiles of NLR, we further transformed NLR from a continuous variable to a categorical variable (Tertile 1:0.00–1.62, Tertile 2:1.62–2.36, Tertile 3:2.36–30.67), and we carried out sensitivity analysis. Participants in the Tertile 3 group experienced an 83% higher risk of coronary heart disease than those in the Tertile 1 group, which was statistically significant (OR = 1.83; 95% CI:1.22–2.73, *P* = 0.0035). Compared to Tertile 1 participants, those in Tertile 2 also had a greater risk of coronary heart disease, although this connection failed to achieve statistical significance (OR = 1.50, 95% CI:0.97–2.31, *P* = 0.0689).

In the fully adjusted model, the chances of coronary heart disease were still substantially influenced by age, sex, hypertension, HDL-C, and LDL-C (Table 4). Compared to male participants, females had 43% reduced chances of coronary heart disease (*P* = 0.0053). Non-hypertensive individuals had a 66% higher risk of coronary heart disease than hypertensive individuals (*P*<0.0001). The probabilities of coronary heart disease decreased by 41% (*P* = 0.0177) and 29% (*P* = 0.0005) for each unit rise in HDL-C and LDL-C, respectively.

As shown in Fig 2A, additional subgroup analysis based on gender, age, smoking status, alcohol use status, BMI, hypertension, and diabetes showed that women (OR = 1.25; 95%

**Table 1. Weighted characteristics of the study population based on NLR tertiles.**

| NLR (1000 cells/uL) | Tertiles 1 (n = 4570) | Tertiles 2 (n = 4572) | Tertiles 3 (n = 4590) | *P* value |
|---|---|---|---|---|
| Age (years) | 58.33 ± 11.12 | 58.60 ± 11.97 | 62.22 ± 12.64 | <0.001 |
| Gender, %(SE) | | | | <0.001 |
| Male | 43.74 | 47.51 | 53.16 | |
| Female | 56.26 | 52.49 | 46.84 | |
| Race, %(SE) | | | | <0.001 |
| Mexican American | 11.47 | 15.05 | 12.22 | |
| Other Hispanic | 10.94 | 10.98 | 10.33 | |
| Non-Hispanic White | 26.02 | 38.91 | 50.92 | |
| Non-Hispanic Black | 34.00 | 19.03 | 14.84 | |
| Other Races | 17.57 | 16.03 | 11.70 | |
| Marital status, %(SE) | | | | <0.001 |
| Married/Living with partner | 60.65 | 63.74 | 60.39 | |
| Widowed/Divorced/Separated | 29.12 | 28.27 | 31.19 | |
| Never married | 10.23 | 7.99 | 8.42 | |
| Education level, %(SE) | | | | 0.084 |
| Less than high school | 11.72 | 12.61 | 11.11 | |
| High school | 36.15 | 34.11 | 35.86 | |
| More than high school | 52.13 | 53.28 | 53.03 | |
| Smoking status, %(SE) | | | | 0.005 |
| Every day | 32.69 | 31.18 | 31.26 | |
| Some days | 7.57 | 7.26 | 5.17 | |
| Not at all | 59.74 | 61.56 | 63.57 | |
| Alcohol use status, %(SE) | | | | <0.001 |
| excessive alcohol use | 16.96 | 17.30 | 20.33 | |
| Moderate alcohol use | 83.04 | 82.70 | 79.67 | |
| BMI, %(SE) | | | | <0.001 |
| 0-25kg/m$^2$ normal | 25.49 | 25.78 | 25.42 | |
| 25-30kg/m$^2$ overweight | 58.20 | 57.23 | 54.94 | |
| >30kg/m$^2$ obese | 16.31 | 16.99 | 19.63 | |
| Hypertension, %(SE) | | | | <0.001 |
| Yes | 46.15 | 45.73 | 54.81 | |
| No | 53.85 | 54.27 | 45.19 | |
| Diabetes, %(SE) | | | | <0.001 |
| Yes | 16.92 | 18.87 | 25.66 | |
| No | 83.08 | 81.13 | 74.34 | |
| SCr (umol/L) | 79.79 ± 34.23 | 80.03 ± 39.50 | 90.09 ± 62.65 | <0.001 |
| SAl (g/L) | 41.85 ± 3.20 | 41.98 ± 3.25 | 41.35 ± 3.55 | <0.001 |
| BUN (mmol/L) | 5.15 ± 2.00 | 5.27 ± 2.02 | 5.90 ± 2.94 | <0.001 |
| HDL-C (mmol/L) | 1.41 ± 0.43 | 1.39 ± 0.42 | 1.37 ± 0.43 | <0.001 |
| LDL-C (mmol/L) | 3.06 ± 0.95 | 3.00 ± 0.94 | 2.79 ± 0.93 | <0.001 |
| Coronary heart disease, %(SE) | | | | <0.001 |
| Yes | 4.25 | 5.03 | 8.67 | |
| No | 95.75 | 94.97 | 91.33 | |

Mean±SD for continuous variables: the *P* value was calculated by the weighted linear regression model.

(%) for categorical variables: the *P* value was calculated by the weighted chi-square test.

Abbreviations: BMI: body mass index; SCr: serum creatinine; SAl: serum albumin; BUN: blood urea nitrogen; HDL-C: high density lipoprotein cholesterol; LDL-C: low density lipoprotein cholesterol; NLR: neutrophil-to-lymphocyte ratio.

**Table 2. Weighted characteristics of the study population based on coronary heart disease.**

| | Coronary heart disease (n = 822) | Non-coronary heart disease (n = 12910) | P-value |
|---|---|---|---|
| Age (years) | 69.42 ± 9.74 | 59.10 ± 11.93 | <0.001 |
| Gender, %(SE) | | | <0.001 |
| Male | 67.15 | 46.93 | |
| Female | 32.85 | 53.07 | |
| Race, %(SE) | | | <0.001 |
| Mexican American | 8.52 | 13.19 | |
| Other Hispanic | 8.15 | 10.91 | |
| Non-Hispanic White | 58.39 | 37.37 | |
| Non-Hispanic Black | 14.60 | 23.12 | |
| Other Races | 10.34 | 15.40 | |
| Marital status, %(SE) | | | <0.001 |
| Married/Living with partner | 58.03 | 61.82 | |
| Widowed/Divorced/Separated | 35.89 | 29.12 | |
| Never married | 6.08 | 9.05 | |
| Education level, %(SE) | | | 0.227 |
| Less than high school | 12.79 | 11.75 | |
| High school | 37.27 | 35.25 | |
| More than high school | 49.94 | 53.00 | |
| Smoking status, %(SE) | | | <0.001 |
| Every day | 24.31 | 32.31 | |
| Some days | 3.75 | 6.81 | |
| Not at all | 71.94 | 60.88 | |
| Alcohol use status, %(SE) | | | <0.001 |
| excessive alcohol use | 23.94 | 17.84 | |
| Moderate alcohol use | 76.06 | 82.16 | |
| BMI, %(SE) | | | 0.072 |
| 0 -25kg/m$^2$ normal | 22.14 | 25.78 | |
| 25-30kg/m$^2$ overweight | 59.12 | 56.65 | |
| >30kg/m$^2$ obese | 18.74 | 17.57 | |
| Hypertension, %(SE) | | | <0.001 |
| Yes | 79.15 | 46.98 | |
| No | 20.85 | 53.02 | |
| Diabetes, %(SE) | | | <0.001 |
| Yes | 41.77 | 19.15 | |
| No | 58.23 | 80.85 | |
| SCr (umol/L) | 101.25 ± 54.77 | 82.14 ± 46.58 | <0.001 |
| SAl (g/L) | 41.03 ± 3.37 | 41.77 ± 3.34 | <0.001 |
| BUN (mmol/L) | 6.87 ± 3.13 | 5.35 ± 2.30 | <0.001 |
| HDL-C (mmol/L) | 1.26 ± 0.39 | 1.40 ± 0.43 | <0.001 |
| LDL-C (mmol/L) | 2.39 ± 0.98 | 2.99 ± 0.93 | <0.001 |
| NLR (1000 cells/uL) | 2.75 ± 1.91 | 2.19 ± 1.27 | <0.001 |

Mean ± SD for continuous variables: the *P*-value was calculated by the weighted linear regression model.

(%) for categorical variables: the *P*-value was calculated by the weighted chi-square test.

Abbreviations: BMI: body mass index; SCr: serum creatinine; SAl: serum albumin; BUN: blood urea nitrogen; HDL-C: high density lipoprotein cholesterol; LDL-C: low density lipoprotein cholesterol; NLR: neutrophil-to-lymphocyte ratio.

Table 3. Associations between NLR and coronary heart disease.

| | OR (95%CI), *P*-value | | |
| --- | --- | --- | --- |
| | Crude model (Model 1) | Minimally adjusted model (Model 2) | Fully adjusted model (Model 3) |
| NLR | 2.15 (1.87, 2.48) <0.0001 | 1.45 (1.25, 1.69) | 1.74 (1.30, 2.33) |
| | | <0.0001 | 0.0002 |
| NLR (Tertiles) | | | |
| Tertiles 1 | Reference | Reference | Reference |
| Tertiles 2 | 1.19 (0.98, 1.45) | 1.04 (0.85, 1.27) | 1.50 (0.97, 2.31) |
| | 0.0744 | 0.7070 | 0.0689 |
| Tertiles 3 | 2.14 (1.79, 2.56) | 1.40 (1.16, 1.69) | 1.83 (1.22, 2.73) |
| | <0.0001 | 0.0004 | 0.0035 |
| *P* for trend | 1.59 (1.44, 1.76) | 1.24 (1.12, 1.38) | 1.37 (1.10, 1.70) |
| | <0.0001 | <0.0001 | 0.0045 |

Notes: NLR(1000 cells/uL) was ln-transferred because of a skewed distribution.

Minimally adjusted for age, gender, race.

Fully adjusted for age, gender, race, Marital status, education level, Smoking status, Alcohol use status, BMI, Hypertension, Diabetes, SCr, SAl, BUN, HDL-C, LDL-C.

Abbreviations: BMI: body mass index; SCr: serum creatinine; SAl: serum albumin; BUN: blood urea nitrogen; HDL-C: high density lipoprotein cholesterol; LDL-C: low density lipoprotein cholesterol; NLR: neutrophil-to-lymphocyte ratio.

CI:1.09–1.43), participants 60 years of age and older (OR = 1.09, 95% CI:1.00–1.19), smoking status for every day or not at all (OR = 1.23, 95% CI:1.00–1.52; OR = 1.09, 95% CI:1.00–1.19), alcohol use status for moderate alcohol use (OR = 1.11, 95% CI:1.00–1.22), BMI>30 kg/m$^2$ (OR = 1.42, 95% CI:1.10–1.82), hypertensive (OR = 1.11, 95% CI:1.02–1.22), and individuals without diabetes (OR = 1.17, 95% CI:1.06–1.31) all showed positive relationships between NLR and coronary heart disease risk. Age, smoking status, alcohol use status, BMI, hypertension, and diabetes did not significantly affect this positive association, according to interaction tests, which did not reveal any statistically significant differences in the relationship between NLR and risk of coronary heart disease (*P* for interaction>0.05).

A smoothed curve fit described the nonlinear association between NLR and coronary heart disease risk (Fig 2B). The smoothed curve fit showed a statistically significant breakpoint of 1.08, which revealed a positive correlation between NLR and coronary heart disease risk (Log-likelihood ratio:0.021, Table 5). Stratified smoothed curve fits revealed a roughly positive link between NLR and coronary heart disease risk in people under 60 years of age and in women (Fig 2C and 2D), a "U-shaped" connection in individuals with hypertension (Fig 2E), and a "U-shaped" relationship with individuals without diabetes (Fig 2F).

## 4. Discussion

Statistics from the NHANES database from 2011 to 2018 were examined for this investigation. We discovered a link between increased NLR and an increased likelihood of coronary heart disease. Subgroup studies and interaction tests revealed comparable findings across populations regarding this connection. Additionally, a "U-shaped" connection, irrespective of diabetes status, was discovered between NLR and risk for coronary heart disease in persons with hypertension.

To the best of our understanding, epidemiological investigations have found a connection between coronary heart disease risk and inflammation levels. In contrast to conventional risk variables, Yang Y-L et al. discovered that SII was a stronger predictor of significant

**Table 4. Multivariate analysis of associations between various variables and coronary heart disease.**

| Variable | OR (95% CI) | *P*-value |
|---|---|---|
| Age (year) | 1.05 (1.03, 1.07) | <0.0001 |
| Gender, %(SE) | | |
| Male | Reference | |
| Female | 0.57 (0.39, 0.85) | 0.0053 |
| Race, %(SE) | | |
| Mexican American | Reference | |
| Other Hispanic | 1.29 (0.61, 2.75) | 0.5015 |
| Non-Hispanic White | 1.58 (0.86, 2.93) | 0.1419 |
| Non-Hispanic Black | 0.63 (0.31, 1.30) | 0.2092 |
| Other Races | 0.95 (0.42, 2.17) | 0.9031 |
| Education level, %(SE) | | |
| Less than high school | Reference | |
| High school | 0.73 (0.43, 1.23) | 0.2370 |
| More than high school | 0.85 (0.50, 1.44) | 0.5350 |
| Marital status, %(SE) | | |
| Married/Living with partner | Reference | |
| Widowed/Divorced/Separated | 0.97 (0.68, 1.39) | 0.8817 |
| Never married | 0.98 (0.51, 1.88) | 0.9432 |
| Smoking status, %(SE) | | |
| Every day | Reference | |
| Some days | 1.26 (0.60, 2.67) | 0.5448 |
| Not at all | 0.78 (0.52, 1.16) | 0.2136 |
| Alcohol use status, %(SE) | | |
| excessive alcohol use | Reference | |
| Moderate alcohol use | 1.41 (0.97, 2.04) | 0.0704 |
| BMI, %(SE) | | |
| 0 -25kg/m$^2$ normal | Reference | |
| 25-30kg/m$^2$ overweight | 0.81 (0.54, 1.20) | 0.2951 |
| >30kg/m$^2$ obese | 0.75 (0.44, 1.29) | 0.3029 |
| Hypertension, %(SE) | | |
| Yes | Reference | |
| No | 0.34 (0.24, 0.50) | <0.0001 |
| Diabetes, %(SE) | | |
| Yes | Reference | |
| No | 0.87 (0.60, 1.25) | 0.4487 |
| SCr (mmol/L) | 1.00 (1.00, 1.00) | 0.4410 |
| SAl (mmol/L) | 0.97 (0.92, 1.01) | 0.1567 |
| BUN (mmol/L) | 1.02 (0.96, 1.09) | 0.4908 |
| HDL-C (mmol/L) | 0.59 (0.39, 0.91) | 0.0177 |
| LDL-C (mmol/L) | 0.71 (0.59, 0.86) | 0.0005 |

Abbreviations: BMI: body mass index; SCr: serum creatinine; SAl: serum albumin; BUN: blood urea nitrogen; HDL-C: high density lipoprotein cholesterol; LDL-C: low density lipoprotein cholesterol.

cardiovascular events in patients with CAD following coronary intervention [22]. Multiple studies have found a strong correlation between CRP levels and the incidence of cardiovascular events, independent of cardiovascular risk or lipid profile, emphasizing inflammation's crucial role in atherosclerotic disease [3, 23–25]. A retrospective investigation by Candemir M

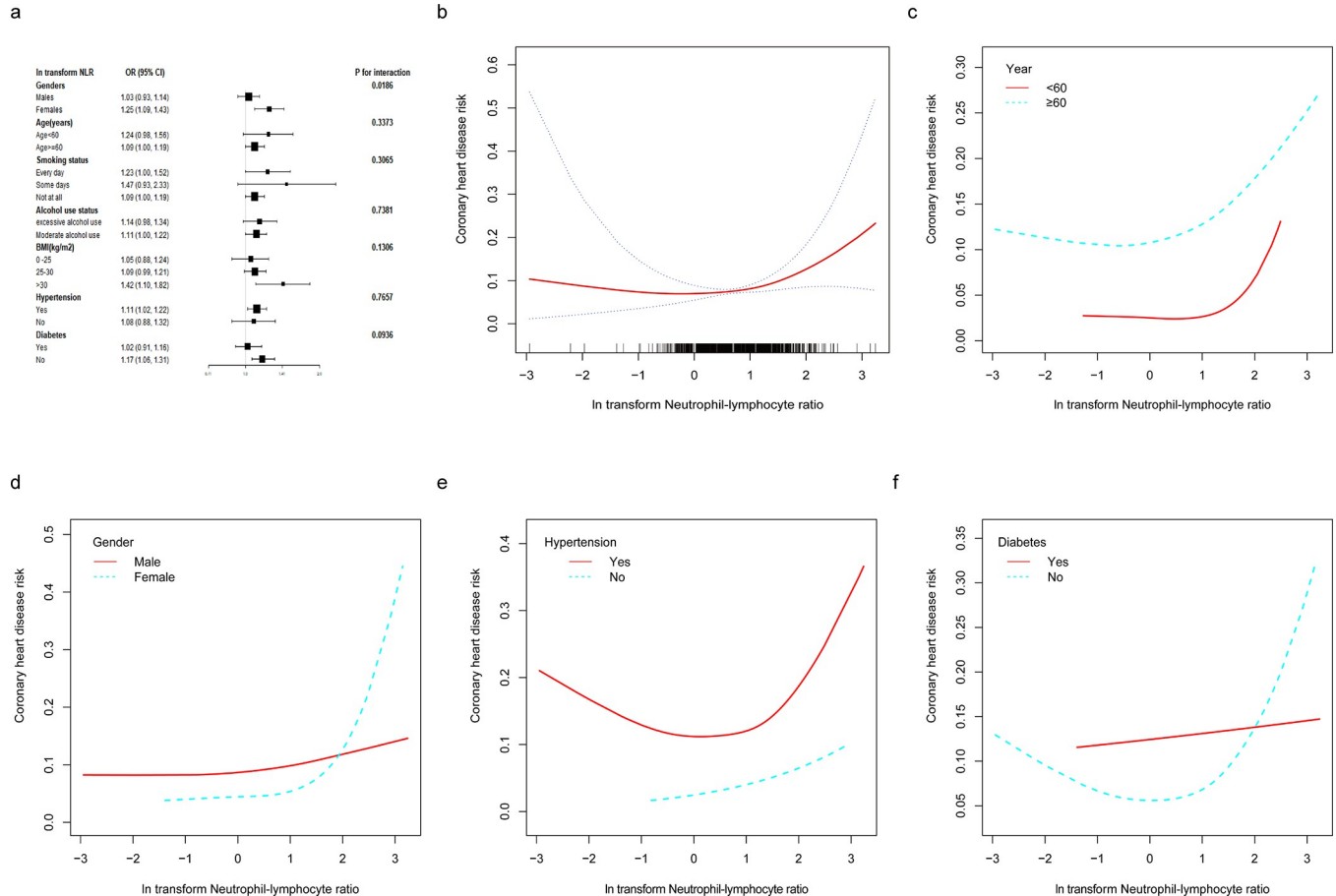

**Fig 2. The association between NLR and coronary heart disease risk.** (a) Subgroup analysis for the association between NLR and coronary heart disease risk. (b) The association between NLR and coronary heart disease risk. The solid red line represents the smooth curve fit between variables. Blue bands represent the 95% confidence interval from the fit. Each black vertical line represents a sample. (c) The association between NLR and coronary heart disease risk stratified by year. (d) The association between NLR and coronary heart disease risk stratified by gender. (e) The association between NLR and coronary heart disease risk stratified by hypertension. (f) The association between NLR and coronary heart disease risk stratified by diabetes.

et al. demonstrated an association between SII levels and coronary heart disease, with a positive correlation between SII and the SYNTAX score, used to measure the severity of coronary atherosclerosis (Rho: 0.630, *P*<0.001) [26]. After multivariate correction, it was discovered in research including 15,758 people that the quantity of abnormally high levels of inflammatory markers was linked to a higher risk of coronary heart disease [27]. Different neutrophil subpopulations were shown to be involved in vascular damage and unstable coronary plaques in SLE, according to Carlucci PM et al. [28]. SII was shown to be a valuable marker to clarify the interplay of thrombocytosis, inflammation, and immunity in the development of vascular disease in a middle-aged and elderly population in a cohort study of 13929 people with a mean age of 62.56 years [29]. In a meta-regression study, the predictive efficacy of inflammatory biomarkers was independent of other confounding variables, and including vascular inflammation biomarkers improved the detection of cardiovascular event risk [30]. Systemic Inflammatory Index and Systemic Inflammatory Response Index were evaluated as new biomarkers of various leukocyte subpopulations in a study evaluating the severity of CAD and its association with the diagnosis of acute coronary syndrome (ACS) or stable CAD in 699 patients. The findings demonstrated an association between SIRI and diagnosis, with the

**Table 5. Threshold effect.**

| Outcome: | Coronary heart disease risk |
|---|---|
| Model I | |
| A straight-line effect | 1.31 (0.97, 1.78) |
| Model II | |
| Fold points (K) | 1.08 |
| < K-segment effect 1 | 0.89 (0.58, 1.37) |
| > K-segment effect 2 | 2.55 (1.36, 4.77) |
| Effect size difference of 2 vs. 1 | 2.85 (1.19, 6.84) |
| Equation predicted values at break points | -2.47 (-2.71, -2.22) |
| Log likelihood ratio tests | 0.021 |

Result variable: Coronary heart disease

Exposure variables: NLR

Results are expressed as OR (95%CI).

Adjusted for age, gender, race, Marital status, education level, Smoking status, Alcohol use status, BMI, Hypertension, Diabetes, SCr, SAl, BUN, HDL-C, LDL-C.

Abbreviations: BMI: body mass index; SCr: serum creatinine; SAl: serum albumin; BUN: blood urea nitrogen; HDL-C: high density lipoprotein cholesterol; LDL-C: low density lipoprotein cholesterol; NLR: neutrophil-to-lymphocyte ratio.

greatest values occurring in patients with ACS (STEMI), which were considerably higher than patients with stable CAD ($P$<0.01), and the highest SII and SIRI values occurring in patients with three-vessel CAD [31]. According to research by Niccoli G et al., effector cells of allergic inflammation play a significant part in the development and instability of coronary artery disease and adverse outcomes following stent placement [32]. Higher SII was substantially linked to an elevated risk of cardiovascular disease, per the findings of 13 studies with 152,996 people included in the analysis (HR = 1.39, 95% CI: 1.20–1.61, $P$<0.001). Elevated SII is linked to an increased risk of cardiovascular disease, suggesting that SII may be a possible biomarker for the emergence of cardiovascular disease [33]. In the crude model (OR = 2.15, 95% CI: 1.87–2.48, $P$<0.0001), the minimally adjusted model (OR = 1.45, 95% CI:1.25–1.69, $P$<0.0001), and the fully adjusted model (OR = 1.74, 95% CI:1.30–2.33, $P$ = 0.0002), we discovered an upward correlation between NLR levels and risk of coronary heart disease. Along with a "U-shaped" relationship between NLR and coronary heart disease risk in participants with hypertension and a "U-shaped" relationship independent of diabetes status, researchers also discovered a threshold effect between NLR levels and coronary heart disease risk, with a breakpoint of 1.08 that was statistically significant (log-likelihood ratio 0.021). In summary, several studies have examined the connection between inflammation and the risk of coronary heart disease. Our results support earlier research that indicated a link between elevated NLR levels and a higher risk of coronary heart disease.

The potential mechanisms behind this beneficial association between inflammation and coronary heart disease risk are not fully understood. Interleukin-1, an inflammatory mediator secreted by neutrophils, may play a role in the abnormal proliferation of vascular smooth muscle cells early in atherosclerosis. This abnormal proliferation of vascular smooth muscle cells can lead to vascular wall degeneration by inducing endogenous PDGF-mediated effects on the proliferation of vascular smooth muscle cells [34, 35]. Lymphocytes, on the other hand, control the inflammatory response and have an anti-atherogenic effect. It has been proposed that in adulthood, the thymus maintains a regulatory population of T cells in peripheral tissues, which prevents early T cells from differentiating into cytotoxic effector cells and encourages the

formation of immune memory, which may have an inhibitory effect on atherosclerosis [36]. Population-based research of chronic coronary artery disease previously discovered the significance of lymphocyte concentration as a predictive marker for coronary artery disease and the relevance of lymphocyte counts in predicting recurrent instability and mortality in individuals with unstable angina [37, 38]. On the other hand, cardiovascular disease risk and prognosis are substantially correlated with C-reactive protein (CRP), one of the indicators of inflammation [39, 40]. Additionally, it is positively correlated with NLR, monocytes, and neutrophils [41]. As a possible substitute for CRP and a sign of systemic inflammation, NLR is a simple marker that may be included in routine practice [42]. NLR and hs-CRP have a positive correlation and may help to detect inflammation [43]. The study mentioned above can show CPR level's direct and indirect effects on NLR's impact on coronary heart disease risk. This could be one of the influence processes.

Our data for this investigation come from the NHANES database, which is well-organized, so-sampled, and differential-rich. First, this study's biggest advantage is the huge number of participants and reliable data. The outcomes were more trustworthy when we adjusted several covariates while analyzing the data and included the vulnerable variables in the multiple regression framework in different formats, such as continuous and categorical variables. False positives are less likely using sensitivity analysis. We also discovered a "U-shaped" link between NLR and coronary heart disease risk in individuals with hypertension and a "U-shaped" relationship with individuals without diabetes after analyzing the nonlinear relationship between NLR and coronary heart disease risk by smoothing curve fitting. Our study does, however, have certain shortcomings. First, there might be some recollection bias as some of the outcome indicators in this study were delivered by questionnaire rather than through objective measurements, and the lack of relevant laboratory test data in the National Health and Nutrition Examination Survey database. Second, because this study is cross-sectional, it is impossible to draw any conclusions about a causal link between NLR and coronary heart disease risk. To further elucidate the causal link between NLR and coronary heart disease risk, follow-up studies require several prospective studies and fundamental research. Even if we took steps to exclude certain confounding variables, other confounding variables may still affect the outcomes. Therefore, follow-up research is required. Our study concluded that there is a link between NLR and a reduced risk of coronary heart disease. Further research is still needed to determine the mechanism underlying it and if reducing NLR levels in clinical settings might minimize the risk of coronary heart disease.

## 5. Conclusions

Our results show a strong correlation between neutrophil-lymphocyte ratio levels and the risk of coronary heart disease. However, the current findings do not demonstrate a causal association, necessitating more thorough prospective research.

## Acknowledgments

We appreciate the data coming from the National Health and Nutrition Examination Surveys.

## Author Contributions

**Conceptualization:** Yu Wang, Yangping Zhuang, Jun Ke.

**Data curation:** Yu Wang, Feng Chen, Jun Ke.

**Formal analysis:** Yu Wang.

**Methodology:** Yangping Zhuang, Jun Ke.

**Project administration:** Changsha Lin.

**Resources:** Changsha Lin.

**Software:** Yangping Zhuang, Changsha Lin, Feng Chen.

**Validation:** Yangping Zhuang, Changsha Lin, Feng Chen, Jun Ke.

**Visualization:** Hanqing Hong.

**Writing – original draft:** Yu Wang, Yangping Zhuang, Changsha Lin.

**Writing – review & editing:** Hanqing Hong, Feng Chen, Jun Ke.

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
