## [Decision Letter · Decision Letter 0]

5 Dec 2023

PONE-D-23-20654The Neutrophil-Lymphocyte Ratio Is Associated With Coronary Heart Disease Risk In Adults: A Population-Based StudyPLOS ONE

Dear Dr. Chen,

Thank you for submitting your manuscript to PLOS ONE. After careful consideration, we feel that it has merit but does not fully meet PLOS ONE’s publication criteria as it currently stands. Therefore, we invite you to submit a revised version of the manuscript that addresses the points raised during the review process.

We look forward to receiving your revised manuscript.

Kind regards,

Aleksandra Klisic

Academic Editor

PLOS ONE

Journal Requirements:

2. Please remove your figures from within your manuscript file, leaving only the individual TIFF/EPS image files, uploaded separately. These will be automatically included in the reviewers’ PDF.

Reviewers' comments:

Reviewer's Responses to Questions

**Comments to the Author**

1. Is the manuscript technically sound, and do the data support the conclusions?

Reviewer #1: Partly

Reviewer #2: No

2. Has the statistical analysis been performed appropriately and rigorously? 

Reviewer #1: No

Reviewer #2: Yes

3. Have the authors made all data underlying the findings in their manuscript fully available?

Reviewer #1: No

Reviewer #2: No

4. Is the manuscript presented in an intelligible fashion and written in standard English?

Reviewer #1: Yes

Reviewer #2: No

5. Review Comments to the Author

Reviewer #1: -The ischemic heart disease in the study is not clear as the questionnaire for ischemic heart disease is not sufficient, the diagnosis of ischemia should be based on non-invasive or invasive highly sensitive test and its results are not identified in addition to presence of established PAD and CVA that increases the risk of CAD

-The laboratory results for risk factors as HBa1C, TG level are missing

Reviewer #2: The authors introduce a cross-sectional study examining the association between neutrophil-to-leukocyte ratio (NLR) and coronary disease in the adult population. The results presented in the manuscript are not groundbreaking but rather confirmatory of prior findings. A significant concern arises from defining the presence of coronary disease solely through participant responses to questions, lacking formal confirmation. Another drawback is the suboptimal reporting quality, potentially hindering reader comprehension. Nevertheless, the topic holds relevance and could contribute to clinical knowledge following substantial revision.

Abstract

“After controlling for relevant confounders, neutrophil-lymphocyte ratio was independently linked to a higher risk of coronary heart disease. A study of the subgroups revealed statistically significant (p<0.05) positive relationships between neutrophil-lymphocyte ratio levels and coronary heart disease risk.”

-Please add point estimates along with confidence intervals

Introduction

Line 40: “The ratios of leukocyte subtypes show the beginning of inflammation”

- Elaborate more on this. Specifically, which ratios are you referring to? This sentence is excessively simplified.

Line 45: “The neutrophil-to-lymphocyte ratio (NLR) is regarded as44 an indicator of inflammation[18-20]. As a result, the connection between NLR and coronary heart disease risk is unclear and needs more research”

- The conclusion reached in the second sentence appears arbitrary and lacks semantic relevance to the previous sentence.

Line 47: “of significant interest in therapeutic therapies”

- Please rephrase

Line 47: “NLR levels must be evaluated concerning coronary heart disease risk”

- This statement lacks coherence.

Material and methods

-Line 77: “survey participants under 16 provided approval from their parents or legal guardians.”

As stated in the last paragraph of introduction, eligible were middle-aged participants (“To ascertain whether there is a correlation between NLR levels and coronary heart disease risk in middle-aged and older populations, this study aimed to investigate the link.”). Hence, subjects aged less than 16 years should have been excluded…

Results

- Line 153: “Participants in our research who had elevated NLR levels were age, male, non-

Hispanic White, smoking use status for not at all, alcohol use status for excessive alcohol use

status, BMI>30 kg/m2, diabetes, serum creatinine, blood urea nitrogen, coronary artery disease.

Non-Hispanic Black and other races had dropped their HDL-C and LDL-C levels”

- This section lacks meaningful content. The input of a native English speaker is essential to enhance the overall flow

-Line 167: “Patients with coronary heart disease risk are more likely to be older, male, non-Hispanic white”

- This information is concisely presented in Table 2 and should be deleted from the main text to improve the reporting quality.

-Line 227: “U-shaped" relationship regardless of diabetes status”

Based on Figure 2f, the relationship appears to be linear for participants with diabetes. Please rephrase

Spell out every abbreviation upon its first use.

An extensive revision of the English language is necessary throughout the manuscript.

6. PLOS authors have the option to publish the peer review history of their article (what does this mean?). If published, this will include your full peer review and any attached files.

Reviewer #1: No

Reviewer #2: No

---

## [Author Response · Author response to Decision Letter 0]

12 Dec 2023

Response Letter

We thank the reviewers for the in-depth review and valuable comments, which have helped us tremendously to improve the quality and clarity of the paper. We have also provided more discussions to strengthen our claims. Please find the point-to-point response to the reviewers’ comments below (Comments from the reviewers are in black, and our responses are highlighted in blue).

Reviewer #1:

We thank reviewer 1 for all suggestions to improve our study. According to the reviewer’s suggestions, we have revised the manuscript and replied to the different points in the following.

Comment 1 and 2:

1.The ischemic heart disease in the study is not clear as the questionnaire for ischemic heart disease is not sufficient, the diagnosis of ischemia should be based on non-invasive or invasive highly sensitive test and its results are not identified in addition to presence of established PAD and CVA that increases the risk of CAD.

2.The laboratory results for risk factors as HBa1C, TG level are missing.

Response 1 and 2:

We thank the reviewers for their valuable comments. Because our data analysis in this manuscript is based on data from the NHANES database, the inability to specify ischemic heart disease and the lack of relevant laboratory test data, as raised by you, the reviewer, is one of our limitations in this manuscript, which we have also explained in the Discussion section of the manuscript (please see lines 325-329). We also reviewed the NHANES database for relevant questionnaires that were administered to respondents by trained interviewers using a computer-assisted personal interview (CAPI) system: Has a doctor or other health professional ever told {you/SP} that {you/s/he} . . .had coronary (kor-o-nare-ee) heart disease? The CAPI system is not only programmed with built-in consistency checking to minimize data entry errors but also uses an online help screen to assist visitors in defining key terms used in the questionnaire so that the quality of the questionnaire can be controlled and ensured. In addition, based on the conclusions from the data analysis of the preliminary database and the reviewers' professional advice and suggestions, our team is now about to carry out a follow-up study and related improvement strategies.

Reviewer #2:

The authors introduce a cross-sectional study examining the association between neutrophil-to-leukocyte ratio (NLR) and coronary disease in the adult population. The results presented in the manuscript are not groundbreaking but rather confirmatory of prior findings. A significant concern arises from defining the presence of coronary disease solely through participant responses to questions, lacking formal confirmation. Another drawback is the suboptimal reporting quality, potentially hindering reader comprehension. Nevertheless, the topic holds relevance and could contribute to clinical knowledge following substantial revision.

We thank reviewer 2 for all suggestions to improve our study. Our data for this study are from a well-designed, well-sampled, and variable-rich NHANES database. The biggest advantage is the large sample size and reliable data. And the results were more reliable when we adjusted different covariates and included the exposure variables in different forms of continuous and categorical variables in the multiple regression model, respectively, in the process of data analysis. As well as sensitivity analysis reduces the possibility of false positives. However, our manuscript also has some corresponding problems, and we have made corresponding revisions to improve the quality of our manuscript according to your professional suggestions and comments as reviewers. Thank you, reviewers, for your hard work.

Comment 1:

Abstract

“After controlling for relevant confounders, neutrophil-lymphocyte ratio was independently linked to a higher risk of coronary heart disease. A study of the subgroups revealed statistically significant (p<0.05) positive relationships between neutrophil-lymphocyte ratio levels and coronary heart disease risk.”

-Please add point estimates along with confidence intervals.

Response 1:

Thank you to the reviewers for your expert advice. We have added point estimates along with confidence intervals to the manuscript; see lines 20-27 of the manuscript.

Comment 2:

Introduction

Line 40: “The ratios of leukocyte subtypes show the beginning of inflammation”

- Elaborate more on this. Specifically, which ratios are you referring to? This sentence is excessively simplified.

Line 45: “The neutrophil-to-lymphocyte ratio (NLR) is regarded as44 an indicator of inflammation[18-20]. As a result, the connection between NLR and coronary heart disease risk is unclear and needs more research”

- The conclusion reached in the second sentence appears arbitrary and lacks semantic relevance to the previous sentence.

Line 47: “of significant interest in therapeutic therapies”

- Please rephrase

Line 47: “NLR levels must be evaluated concerning coronary heart disease risk”

- This statement lacks coherence.

Response 2:

Thank you, reviewer, for your expert guidance. We have made corresponding changes according to your suggestions; please see lines 45-48 and 53-56 of the manuscript.

Comment 3:

Material and methods

-Line 77: “survey participants under 16 provided approval from their parents or legal guardians.”

As stated in the last paragraph of introduction, eligible were middle-aged participants (“To ascertain whether there is a correlation between NLR levels and coronary heart disease risk in middle-aged and older populations, this study aimed to investigate the link.”). Hence, subjects aged less than 16 years should have been excluded…

Response 3:

Thanks to the reviewers for their meticulous work. I apologize for our oversight, which was an oversight in our writing. We excluded subjects under 16 years of age, as seen in our flowchart of the manuscript(Fig1).

Comment 4:

Results

- Line 153: “Participants in our research who had elevated NLR levels were age, male, non-

Hispanic White, smoking use status for not at all, alcohol use status for excessive alcohol use

status, BMI>30 kg/m2, diabetes, serum creatinine, blood urea nitrogen, coronary artery disease.

Non-Hispanic Black and other races had dropped their HDL-C and LDL-C levels”

- This section lacks meaningful content. The input of a native English speaker is essential to enhance the overall flow

-Line 167: “Patients with coronary heart disease risk are more likely to be older, male, non-Hispanic white”

- This information is concisely presented in Table 2 and should be deleted from the main text to improve the reporting quality.

-Line 227: “U-shaped" relationship regardless of diabetes status”

Based on Figure 2f, the relationship appears to be linear for participants with diabetes. Please rephrase

Response 4:

Thank you, reviewers, for your hard work in reviewing the manuscript. We have made relevant deletions and revisions based on your professional and careful guidance. Details can be found in lines 159-163, and 228-229 of the manuscript.

Comment 5:

Spell out every abbreviation upon its first use.

An extensive revision of the English language is necessary throughout the manuscript.

Response 5:

Thank you, reviewers, for your expert guidance. We have fully spelled out the abbreviations used for the first time as well as extensively revised the English language throughout the manuscript. For example, in lines 17, 30, 59, etc. of the manuscript.

In conclusion, we highly appreciate the rigorous and detailed professional questions raised by the reviewers, which helped our team optimize the details more scientifically rigorously to support our viewpoints and prove the significance of this study. We sincerely hope that the detailed and meticulous explanations and analyses herein will shed light on the scientific rigor of this study. We sincerely hope the reviewers will appreciate our explanations in the revised manuscript.

---

## [Decision Letter · Decision Letter 1]

20 Dec 2023

The neutrophil-to-lymphocyte ratio is associated with coronary heart disease risk in adults: A population-based study

PONE-D-23-20654R1

Dear Dr. Chen,

We’re pleased to inform you that your manuscript has been judged scientifically suitable for publication and will be formally accepted for publication once it meets all outstanding technical requirements.

Kind regards,

Aleksandra Klisic

Academic Editor

PLOS ONE

Additional Editor Comments (optional):

Reviewers' comments:

Reviewer's Responses to Questions

**Comments to the Author**

1. If the authors have adequately addressed your comments raised in a previous round of review and you feel that this manuscript is now acceptable for publication, you may indicate that here to bypass the “Comments to the Author” section, enter your conflict of interest statement in the “Confidential to Editor” section, and submit your "Accept" recommendation.

Reviewer #1: All comments have been addressed

Reviewer #2: All comments have been addressed

2. Is the manuscript technically sound, and do the data support the conclusions?

Reviewer #1: Partly

Reviewer #2: Yes

3. Has the statistical analysis been performed appropriately and rigorously? 

Reviewer #1: Yes

Reviewer #2: Yes

4. Have the authors made all data underlying the findings in their manuscript fully available?

Reviewer #1: Yes

Reviewer #2: Yes

5. Is the manuscript presented in an intelligible fashion and written in standard English?

Reviewer #1: Yes

Reviewer #2: Yes

6. Review Comments to the Author

Reviewer #1: Ti accept the manuscript he neutrophil-to-lymphocyte ratio is associated with coronary heart disease risk in

adults: A population-based study for publication.

Reviewer #2: Thank you for the opportunity to review this interesting study. My comments have been addressed appropriately.

7. PLOS authors have the option to publish the peer review history of their article (what does this mean?). If published, this will include your full peer review and any attached files.

Reviewer #1: No

Reviewer #2: No

---

## [Editor Report · Acceptance letter]

2 Feb 2024

PONE-D-23-20654R1 

PLOS ONE

Dear Dr. Chen, 

I'm pleased to inform you that your manuscript has been deemed suitable for publication in PLOS ONE. Congratulations! Your manuscript is now being handed over to our production team.

Kind regards, 

on behalf of

Dr. Aleksandra Klisic 

Academic Editor

PLOS ONE